# Evolutionary Trends in the Mitochondrial Genome of Archaeplastida: How Does the GC Bias Affect the Transition from Water to Land?

**DOI:** 10.3390/plants9030358

**Published:** 2020-03-12

**Authors:** Joan Pedrola-Monfort, David Lázaro-Gimeno, Carlos G. Boluda, Laia Pedrola, Alfonso Garmendia, Carla Soler, Jose M. Soriano

**Affiliations:** 1Cavanilles Institute of Biodiversity and Evolutionary Biology, University of Valencia, 46980 Paterna, Spain; joan.pedrola@uv.es (J.P.-M.); david.lazaro-gimeno@uv.es (D.L.-G.); Carlos.Boluda@ville-ge.ch (C.G.B.); laiapedrolavidal@gmail.com (L.P.); 2Unité de Phylogénie et Génetique Moléculaires, Conservatoire et Jardin Botaniques, Chambésy, 1292 Geneva, Switzerland; 3Mediterranean Agroforestry Institute, Department of Agroforest Ecosystems, Polytechnic University of Valencia, 46022 Valencia, Spain; algarsal@upvnet.upv.es; 4Biomaterials, Institute of Materials Science, University of Valencia, 46980 Paterna, Spain; carla.soler@uv.es

**Keywords:** Archaeplastida, GC bias, equilibrium GC frequency, GC content concomitance, mitochondrial genomic pattern

## Abstract

Among the most intriguing mysteries in the evolutionary biology of photosynthetic organisms are the genesis and consequences of the dramatic increase in the mitochondrial and nuclear genome sizes, together with the concomitant evolution of the three genetic compartments, particularly during the transition from water to land. To clarify the evolutionary trends in the mitochondrial genome of Archaeplastida, we analyzed the sequences from 37 complete genomes. Therefore, we utilized mitochondrial, plastidial and nuclear ribosomal DNA molecular markers on 100 species of Streptophyta for each subunit. Hierarchical models of sequence evolution were fitted to test the heterogeneity in the base composition. The best resulting phylogenies were used for reconstructing the ancestral Guanine-Cytosine (GC) content and equilibrium GC frequency (GC*) using non-homogeneous and non-stationary models fitted with a maximum likelihood approach. The mitochondrial genome length was strongly related to repetitive sequences across Archaeplastida evolution; however, the length seemed not to be linked to the other studied variables, as different lineages showed diverse evolutionary patterns. In contrast, Streptophyta exhibited a powerful positive relationship between the GC content, non-coding DNA, and repetitive sequences, while the evolution of Chlorophyta reflected a strong positive linear relationship between the genome length and the number of genes.

## 1. Introduction

Mitochondrial, plastidial and nuclear genome lengths (GL) increased dramatically due to the addition of non-coding DNA (%NC) during the evolution of green plants, particularly during the transition from water to terrestrial life. This phenomenon occurred in parallel with an increase in Guanine-Cytosine (GC) content (%GC) and organism complexity [1]. The interactions among GL, %NC, the number of repeated sequences (NRS), their total length (RSL), and the %GC are not yet well understood for any of the three genetic compartments; however, plastids appear to be the most evolutionarily stable, with few changes in GL or %GC [2]. Evolutionary changes in the aforementioned variables may have occurred concurrently at two or three genetic compartments, and the factors determining this concurrence or the lack thereof are among the most intriguing puzzles in the evolutionary biology of primary photosynthetic eukaryotes.

Archaeplastida include Glaucophyta, Rhodophyta (red algae), and Viridiplantae (green plants), although the monophyly of this group is not exempt from controversy [3,4]. Green plants are further divided into two main clades: Chlorophyta, including most unicellular and marine algae; Streptophyta, including most freshwater algae and land plants [5]. All these lineages have three genetic compartments with well-coordinated working biochemical machinery, and they differ significantly in their architecture and evolution [6]. We can see that lateral transfer among the three compartments occurred, especially from the organelles to the nucleus and from the plastid to the mitochondrion, whereas transfer from the mitochondrion to the plastid was almost non-existent [7]. Meaningful differences can be reflected in some of the mitochondrial genome (mtDNA) characteristics in the Archaeplastida lineages, such as the GLs, genetic code, codon usage, gene content, and the degree of ribosomal gene fragmentation [8]. Throughout the transition from water to land, terrestrial plants acquired some peculiar features in their mtDNA, including large genomes with a high %NC, editing at the transcriptional level, genomic recombination, trans-splicing introns, foreign DNA insertions, lateral gene transfer, and gene duplications. These features are not yet widely studied in streptophyte green algae and early land plants [9]. Selection was tested in both Viridiplantae lineages as the driving force that increases mtDNA and %GC. Two very different patterns arose: strong selection likely affected the codon usage in Chlorophyta and mutation, and genetic drift appeared to be the major evolutionary driving forces for Streptophyta [10,11]. 

These patterns in Streptophyta mtDNA, produced essentially by non-adaptive forces, depended on the effective population size, generation times and the differences between unicellularity and multicellularity, which was consistent with the previous findings in plastids [12]. If strong selection was excluded, the challenge was to determine which other evolutionary force can explain the %GC increases throughout Streptophyta. Other possible explanations proposed for the GC bias, apart from selection, were the mutational bias and GC-biased gene conversion (gBGC) [13,14,15]. The ribosomal DNA (rDNA) genes are among the most conserved sequences in the three genetic compartments and have very specific evolutionary dynamics, including their long-term high recombination rate due to a concerted evolution. 

They are also some of the most widely available sequences from eukaryotes in genomic databases. Therefore, they were considered useful genetic markers when analyzing the %GC variation throughout the entire genome. Based on the rDNA polymorphism data, the variation in nuclear rDNA %GC throughout the phylogenetic trees of angiosperms and vertebrates was observed [15] with a strong SNP (single nucleotide polymorphism) excess, for which either G or C was the majority allele. This was inconsistent with the mutational bias hypothesis and supported the GC-biased gene conversion (gBGC)/selection-driven evolution hypothesis. 

The most outstanding aspect of this biased gene conversion was its impact on %GC, which affect the functional components of the genome and impeded natural selection (the Achilles’ heel hypothesis) [14]. The gBGC appeared to play a significant role in the evolution of the genetic systems (e.g., sexual reproduction and recombination, inbreeding avoidance mechanisms, and ploidy cycles) and the development of the senescence and degeneration of non-recombining regions [16]. Despite the importance of this evolutionary force, the phylogenetic gBGC distribution in Streptophyta was not yet studied in association with the transition from water to land at the three genetic compartments.

Is Archaeplastida mitochondrial %GC related to the GL, %NC, NRS, gene number (GN) or coding sequences? If so, how are they linked and what role does this play in different lineages? Is there heterogeneity in the base composition through the streptophytes phylogeny? How is the %GC distributed in the three genetic compartments throughout the Streptophyta tree? Is the %GC increase concomitant for the three genome compartments? To answer these questions, our aims were the following: (i) To analyze the genomic variables with phylogenetically independent linear models in order to depict the evolutionary patterns of the mitochondrial genome in Archaeplastida. (ii) To examine the heterogeneity in base composition among the branches of the Streptophyta tree, using non-homogeneous models of sequence evolution, taking into account the phylogenetic relationships. (iii) To implement a reconstruction of the ancestral GC content and GC* (equilibrium GC frequency) [17] in the three genetic compartments throughout the Streptophyta phylogenetic tree to compare their evolution.

## 2. Results

### 2.1. Genome Features

Across Archaeplastida evolution (Figure 1), taking all clades and studied variables into account (Appendix A), the only significant relationship found was the one between the NRS and GL, with both logarithms transformed for linearity (Appendix A). These two variables were also linearly related to %NC; however, these relationships became unclear when the Streptophyta lineage was removed from the analyses, and became more significant when the Chlorophyta lineage was the one excluded (Appendix A). 

Excluding the Chlorophyta lineage, the analyses determined an evident increase of the GL, %GC, %NC and NRS, across the Streptophyta lineage. The %GC exhibited a strong significant linear relationship with the %NC and log (NRS) (Figure 2a,b; Appendix A). The effect of this last variable was mediated by the %NC (Figure 2c), and therefore, it lost its significance when the %NC effect was accounted for. In other words, the NRS affected the %NC and this explained 46% of the variance for the %GC. The GL was not directly related to the %GC, however it was clearly affected by the %NC, which explains 82 % of its variance (Figure 2d). The effect of the NRS over the GL was also mediated by the %NC. On the other hand, when the Streptophyta clade was excluded from the analyses, the GL appeared to be related directly to the number of protein-coding genes (NPG) (Appendix A).

Meanwhile, across Archaeplastida, the number of genes ranged greatly, both with lineages and between them. The main pattern was the gene number reduction in the Chlorophyceae family, along with an extreme reduction in the mitochondrial genome. The genes maintained in most species were those for tRNA, rRNA and ribosomal proteins and those involved in respiration and oxidative phosphorylation (Appendix A). Of these, only six genes were preserved in the mtDNA in all the studied species (cob, cox1, nad1, nad4, nad5 and nad6). Therefore, mitochondria lost most of the original bacterial genes and conserved only those associated with their principal cellular functions, respiration and oxidative phosphorylation, and those involved in the genetic machinery (rRNA and tRNA), but not in all cases. Two hornworts (Nothoceros aenigmaticus and Phaeoceros laevis) lost most of the ribosomal protein-coding genes [18,19], as did *Selaginella*, which also lost all tRNA genes [20]. 

In addition, Chlorophyceae did not maintain any genes for ribosomal proteins, and the tRNAs were reduced considerably (Scenedesmus obliquus retained more tRNA genes but lost all ribosomal genes: rRNA and ribosomal protein-coding genes).

### 2.2. Heterogeneity in the Base Composition of Ribosomal Subunits in Streptophyta and the Reconstruction of the Ancestral %GC and GC* Content

The likelihood ratio test of non-homogeneous models showed the heterogeneity in the base composition among the clades studied. The non-homogeneous “terminal clades” hierarchical model fit the sequence heterogeneity better for the mtLSU, the cpLSU and the cpSSU (Table 1). Through the evolution of the streptophytes, from unicellular algae to angiosperms, through charophytes, mosses, ferns, and other lineages, the mtSSU increased its %GC. However, this was not in a progressive manner, but was variable according to the clades (Figure 3).

This process exhibited a clear pattern for the mtSSU, with differences among the seed plants, ferns, and club mosses, as well as the other lineages. The ribosomal nSSU resembled this augmentation pattern, but it was much less clear for the cpSSU. In all cases, the mtSSU contained noticeably less %GC than the cpSSU and the nSSU. The nSSU %GC ranged from 57.16% in the Coleochaetophyceae to 62.17% in the Angiosperms and, regarding the mtSSU, from 45.24% in the Coleochaetophyceae to 51.36% in the Angiosperms (a 6% increase). The cpSSU also demonstrated a %GC augmentation pattern, ranging from the Klebsormidiophyta at 57.28% to 65.46% in the Monilophyta (an 8.3% increase), with a high %GC for the Zygnematophyceae (63.37%) and a low %GC for the Charophyceae (58.49%) and Coleochaetophyceae (59.64%). Although not concurrently, these patterns are similar for the LSU, for both organelles studied (Appendix A).

A notable result was the low GC* in the mitochondrial Anthocerotophyta ribosomal subunits (19.04%). This very low value seemed to be unrelated to any of the biological or genomic characteristics studied; thus, we had no explanation for this phenomenon, despite the peculiarly high amount of editing in this phylum. The large %GC and GC* in the Charophyceae nSSU were also surprising. This pattern was maintained when the analyses were repeated without invariant sites. 

Furthermore, we thoroughly looked for concomitancy of the patterns of the changes in the %GC, between the genetic compartments associated with the species and clades. We did not find any clear relation apart from the one between the large and small plastidial subunits, and only when coincident species were used. Therefore, we can ensure that although being present in the three genetic compartments, the increase in %GC followed a distinct pattern for each one, not concurrently through the lineages.

## 3. Discussion

### 3.1. Archaeplastida Mitochondrial Trends

The Archaeplastida mitochondrial genomes demonstrated evidence that the evolution of these genomes followed differentiated paths between the major lineages, as the evolutionary process is unpredictable due to many chance events. In the framework of population genetics theory, the mutational-hazard hypothesis predicted a favorable environment for the proliferation of non-coding mtDNA with a low product between the effective gene number per locus in the population (Ng) and the mutation rate (μ). The small Ngμ value in the mtDNA of green plants (Ngμ << 1) compared to animals (Ngμ >> 1) intensified the genetic drift, making it easier for alleles with high mutation rates to behave neutrally, and thereby encouraging their fixation in the population [21]. This phenomenon could explain the difference between the small size and high mutation rate of animal mitochondrial genomes and the dramatic accumulation of non-coding mtDNA and the low mutation rate through the evolution of green plants. 

In Streptophyta, an increase in the NRS incremented the %NC, which in turn facilitated the recombination and consequently raised the %GC through biased gene conversion. The best explanations for this observed gain in the GL and %GC appeared to be the mutational-hazard and the GC-biased gene conversion (gBGC) hypotheses, respectively. The promoted recombination was followed by an increase in %GC through biased gene conversion [13,21]. 

These non-adaptive forces occurring along the transition from water to land represented an exaptive genomic platform that explained some of the observed directional trends in the evolution of the land plant genomes: the recombination extension and the expansion of the genomic regulatory areas provoked a progressive accumulation of certain protein families correlated with the cell type number, which was essentially caused by gene duplication, concomitantly with the increase in organism complexity [22]. 

The chlorophytes are generally unicellular and sometimes parasites. They exhibit a GL reduction when the gene number decreases. By contrast, the Streptophyta evolution led to multicellular land plants, with a combination of striking features: organismal complexity together with a dramatic increase of the GL and %GC on the mitochondria [23].

The obtained models from our results were very consistent despite the relatively low taxon sampling. These observations supported the hypothesis that the changes in these variables were caused by the same factors. Thus, genome enlargement could provide a favorable environment for an increase in the recombination rate; therefore, a greater number of mismatches (that should be repaired by specific enzymes) would be produced, raising the probability for the incorrect nucleotides being replaced by G or C via gene conversion bias. Consequently, the %GC could be used as a fingerprint for the amount of recombination in these genomes. These mtDNA recombination mechanisms (surveillance and repair recombination-induced DNA damage, including mismatch repair) probably needed the selective pressure for efficient repair to be relaxed, raising the mutation rate [13]. The mitochondrial complexity augmentation throughout evolution did not occur in the mtDNA of multicellular animal lineages, although it did in their nuclear genome [24]. 

In Archaeplastida, the three genetic compartments must interact in coordination; consequently, a strong interactive adaptation between the organelles and the nucleus was necessary during evolution to multicellularity [25]. Hence, the three genomes increased their biological complexity in land plants, which permitted them to perform different functions in different tissues. Generally, species with a high %GC cpDNA also have %GC mtDNA but there are many exceptions [26,27]. In fact, evolutionary factors acting on the organism are the same in all three genetic compartments and consequently the differences indicated their variate idiosyncrasy and origin. 

### 3.2. The Distribution of GC Content and GC* in the Three Genetic Compartments for Streptophyta

Basal Streptophyta algae evidenced a different pattern for %GC in their genetic compartments than Charophyceae and the terrestrial lineages. The %GC increased, especially from early land plants to angiosperms, yet there were intriguing exceptions to this pattern, such as the Charophyceae high %GC and GC* on the nSSU and mtSSU and the low value on cpSSU.

In the most basal streptophytes algae, the disparity in the %GC between the ribosomal subunits of the three genetic compartments resulted from strong selection, rather than from the gBGC, following a similar pattern to that observed in Chlorophyta [28]. Some aspects of the population biology of these lineages may have also been indirectly responsible for the %GC, because, as mentioned above, the population size and mutation rate determined that GL. *Chlamydomonas* is a Chlorophyceae algae living in freshwater habitats, with a similar ecology to Mesostigma; therefore, these genera are expected to have similar, very large populations. 

The Streptophyta algae are multicellular (except for the Desmidiales), though very small (except for the Charophyceae), and, consequently, they may maintain large populations. In contrast, the Charophyceae have a large body size, with few individuals living in ponds or lagoons, and, hence, a very small effective population size, although they are multinucleated. Terrestrial plants also have small populations compared to those of the basal Streptophyta algae. In fact, Harholt et al. [29] hypothesized that streptophyte algae lived on land before the emergence of embryophytes, which was verified, recently, by Wang et al. [3], in the common ancestor of *Mesostigma viride* and *Chlorokybus atmophyticus*, where the development of traits reflected adaptations to a subaerial/terrestrial habitat.

The reasons for the differences in the recombinations and the gBGC among genomes, species and lineages remains unknown. Nevertheless, in the Charophyceae, these high %GC and GC* in the nucleus can be explained by the large C value (2C nuclear DNA content) (10–50 pg), which was higher than for the rest of the streptophytes algae (0.7–5 pg) and all mosses studied (0.9–5 pg) [30].

## 4. Materials and Methods 

### 4.1. Archaeplastida Mitochondrial Genome-Wide Characteristics

All 35 Archaeplastida mitochondrial genomes present in the NCBI database were selected. Two outgroups were also chosen for their unique characteristics. The protozoan *Reclinomonas americana* (Excavata) was the shortest mitochondrial genome from eukaryotes and probably the one that best reflected the ancestral state [31]. Additionally, the parasite *Rickettsia prowazekii* (α-proteobacteria) is one of the closer derivatives from the eubacterial organisms, where the mitochondrial organelles originated [32]. The samples represented the major Archaeplastida clades, including Glaucophyta (*n* = 2), Rhodophyta (*n* = 4), Prasinophyta (*n* = 4), basal Chlorophyta (*n* = 5), Chlorophyceae (*n* = 6), basal Streptophyta (*n* = 3), Charophyceae (*n* = 2), including *Nitella hyalina* (GenBank database code JF810595), basal Embryophyta (*n* = 5), and Spermatophyta (*n* = 4). Certain features of the genome, including the %GC, were extracted from the databases at NCBI, using Artemis software [33]. The following variables were also obtained from all species: GL, %GC, %NC, NPG, NRS and RSL. The complete list of species, accession numbers and genome characteristics are available in the Appendix A. 

To find the repeated sequences, the complete mtDNA sequences (forward, reverse, complement and reverse complement) from selected species were analyzed with REPuter [34]. The minimal repeat size was limited to 20 nucleotides (nt) for general analyses, 50 nt for less repetitive mitochondrial sequences and 100 nt in more repetitive sequences, when using the largest repeats in each genome. The %GC within repetitions was determined using Emboss [35].

Linear model analyses were used to determine the shape and significance of the relationship between each pair of variables. These linear models were implemented in three different ways to test for consistency: The classic linear regression and two different phylogenetically independent contrasts: “the Phylogenetic Generalized Least Squares (PGLS) method and “crunch” method, both from the “caper” package in R [36]. PGLS was a powerful method to estimate the adaptive optima using continuous data [37]. This method assumed that the analyzed trait evolved by Brownian motion and thus trait covariance between any pair of taxa decreased linearly with time (branch length) since their divergence. The methods were originally provided in the programs CAIC [38] and MacroCAIC [39]. Both programs calculated phylogenetically independent contrasts in a set of variables and then used linear models of those contrasts to test for evolutionary relationships. All contrast model functions enforced regression through the origin. Mediation tests were also performed to determine whether the relationships between pairs of variables were mediated by a third variable.

All these analyses were carried out on three different data sets: (i) all the studied species without the outgroups, (ii) eliminating the Chlorophyta, and (iii) eliminating the Streptophyta. These two lineages evolved by different evolutionary paths, and therefore represented different non-comparable patterns.

The assumptions of normality and homoscedasticity of the residuals were evaluated to verify the appropriateness of the linear modeling. A Bonferroni correction was carried out to correct the results from all these multiple tests and therefore, linear relationships were considered significant only when *p* < 0.005.

The phylogenetic tree used for the independent contrasts was built from the only six protein-coding genes that were shared by the 37 species studied (cob, cox-1, nad1, nad4, nad5 and nad6). These genes sequences were translated into amino acids with TranslatorX [40], aligned with Muscle [41] and trimmed with GBLOCKs [42] with the default parameters, resulting in a concatenated matrix with 1725 amino acids. The sequences matrix for each gene was subjected to ProtTest to find the best fit evolutionary model [43]. In order to test the phylogenetic signal TREE-PUZZLE was used [44]. For the maximum-likelihood (ML) analyses, the concatenated protein matrix was analyzed with RAxML v. 7.2.8 [45] using the WAG model [46] and a bootstrap analysis with 1000 replicates. 

Bayesian analyses were implemented with Mr. Bayes V.2.1.0 [47]. The concatenated protein matrix was analyzed using three model partitions: two fixed cpREV models [48] of amino acid substitution, with inv-gamma and gamma distributions of rates, and the third model with a fixed Jones model with gamma distribution of rates. The analyses, in all cases, consisted of three million generations, four independent runs and four Markov chains. The trees were sampled every 1000 generations; stationarity was assessed by examining the standard deviation of the split frequencies and by plotting the –ln L per generation using Tracer v1.4 [49], and the trees were generated before the stationarity was discarded. Further Bayesian analysis with a fixed-cpREV model for the six coincident mitochondrial protein-coding genes was consistent with the tree obtained with ML and proved to be the best-fitting tree to most accepted phylogenies. Therefore, this tree was utilized for the statistical phylogenetically independent analyses.

### 4.2. Analyses of Heterogeneity in Base Composition

Mitochondrial ribosomal small and large subunits (mtSSU and mtLSU), chloroplast ribosomal subunits (cpSSU and cpLSU), and nuclear ribosomal small subunit (nSSU) sequences were downloaded from the SILVA database [50]. A hundred species for each ribosomal subunit were selected in order to cover all available lineages across the Streptophyta tree, discarding short, incomplete or highly gapped sequences. The species names for each subunit and genetic compartment are available in the Appendix A. 

Zygnematophyceae were not represented in the mitochondrial subunits, nor were Klebsormidiophyceae in the mtSSU, as there were no available sequences in the databases. Homologous rRNAs for each subunit and genetic compartment were aligned using ClustalW [51]. Escobar et al. [15] eliminated the hypermutable CpG sites to prevent slippage in the angiosperm %GC calculation. However, as Archaeplastida lineages diverged very early, hypermutable CpG site detection cannot be performed in a reliable way, so this step was omitted from the analyses [52]. The length of the sequences used had approximated sizes of 1289 (mtSSU), 1384 (mtLSU), 1272 (nSSU), 1162 (cpSSU) and 2155 (cpLSU) base pairs. 

Phylogenetic trees were inferred with PhyML.3.0 [53] in the five RNA markers used. The models of sequence evolution were obtained with the program JModeltest2 [54] using the general time reversible model (GTR) + I + G, with four categories for the gamma distribution, parsimony starting trees and SPR (sub-tree pruning and regrafting) branch swapping. In general, terminal clades were grouped consistently with the most current Streptophyta phylogeny, although deep clades were less resolved. Therefore, some branches of the trees resulting from the above analyses were relocated with Baobab software [55], in order to adjust them to the accepted phylogenies [56]. We thensubsequently re-optimize the branch length with PhyML. The ML analyses of non-homogeneous models presented here were implemented with the modified trees. However, analyses with unmodified trees were performed to test for robustness. 

The heterogeneity in %GC was tested with four non-homogeneous models of sequence evolution. These models were fitted with the software BppML [57] and NHML [58], which utilize an ML approach that is not stationary (ancestral and current %GC may differ) and with no homogeneity (the branches may have different %GC) in the base composition across the phylogeny. Thus, the %GC at the nodes and the GC* at branches of the phylogenetic tree were estimated. These hierarchical models were fitted to test whether the branches underwent similar evolution in their base composition or not, using the fixed trees from PhyML. Nucleotide substitution models based on Galtier and Gouy [58] were implemented for all BppML analyses, using four gamma categories and two parameters: theta (GC*) and kappa (Ts/Tv). 

The hierarchical models used were: (1) a homogeneous model; (2) a non-homogeneous model (M1), Angiosperms-Gymnosperms (seed plants) + Monilophyta-Lycopodiophyta + Bryophyta-Marchantiophyta + Anthocerotophyta + Charophyceae + Coleochaetophyceae + Zygnematophyceae + Klebsormidiophyceae + Chlorokybophyceae + Mesostigma; (3) a non-homogeneous model (M2), which was the same as above but splitting Marchantiophyta and Bryophyta; (4) a non-homogeneous model of terminal clades; and (5) a non-homogeneous model. One was used per branch, where each branch had its own %GC and GC*. The likelihood ratio test (LRT) was used to assess whether more complex nested models provided a significantly improved fit compared with simpler models.

### 4.3. Streptophyta Ribosomal GC Content and GC*

The %GC and GC* were estimated for all species and nodes across the Streptophyta phylogenetic tree, using the same method that Escobar et al. [15] used. Therefore, the GC* was defined as:(1)$$GC=\frac{AT\to GC}{AT\to GC+GC\to AT}$$
where AT→GC refers to the substitution rate from A or T to G or C bases and GC→AT holds for the inverse. The GC* was considered a more appropriate estimator for evolutionary dynamics than the %GC, because GC* reflects the relative contribution of changes from AT to GC independently from the total number of mutations [59]. NHML software [58] was used to implement a ML approach for the non-stationary probabilistic and non-homogeneous model. The objective set was to reconstruct the ancestral %GC and GC* distributions optimizing parameters on the model “terminal clades” over the trees obtained from BppML, for each one of the five ribosomal subunits. The analyses were performed with and without invariable sites to check the consistency. Sequence gaps were removed in all cases. The results from these analyses were two phylogenetic trees for each of the ribosomal subunits analyzed, with pseudo-bootstrap values for %GC and GC*. The GC* is the theta parameter estimated in this ML framework [58].

### 4.4. Concomitant Evolution between Genetic Compartments and Clades

With the aforementioned trees, phylogenetically independent contrasts (PGLS and “crunch”) of %GC were carried out to check for concomitant evolution. The following contrasts were made: between the small and the large subunits for each genetic compartment; between the large subunits of the mitochondrion and plastid; between the small subunits of the three different genetic compartments. These tests were performed in two different ways: in the first instance, using the %GC in the shared species between subunits pairs (Appendix A), and second, using the node values (NHML) in the Streptophyta clades, despite the species not being the same. In both cases, each analysis was implemented with three different tree topologies: the phylogenetic tree of the two subunits studied and the tree with a fixed topology (all branch lengths equal to one). These analyses were also conducted with and without invariant sites and with the Bonferroni corrected for multiple (six) comparisons, considering significant differences only when *p* < 0.008.

## Figures and Tables

**Figure 1 plants-09-00358-f001:**
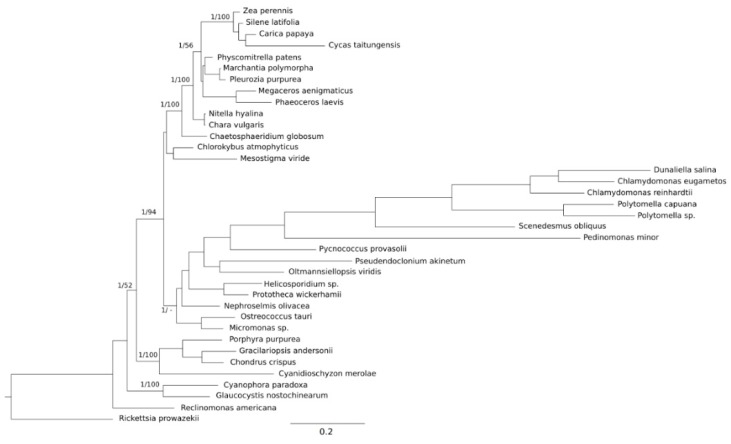
The phylogenetic relationships of the main clades of Archaeplastida, determined by six concatenated mtDNA-encoded proteins (cob, cox1, nad1, nad4, nad5, nad6), shared between the 37 studied genomes. The phylogenetic analyses were summarized by a Bayesian inference (BI) (cpREV model) consensus tree, with branch support from both BI and maximum likelihood (ML) analyses (Bayesian posterior probabilities/ML bootstrap). The prokaryote *Rickettsia prowazekii* and the protozoan excavate *Reclinomonas americana* were used as the outgroups. The scale bar represents the estimated number of amino acid substitutions per site.

**Figure 2 plants-09-00358-f002:**
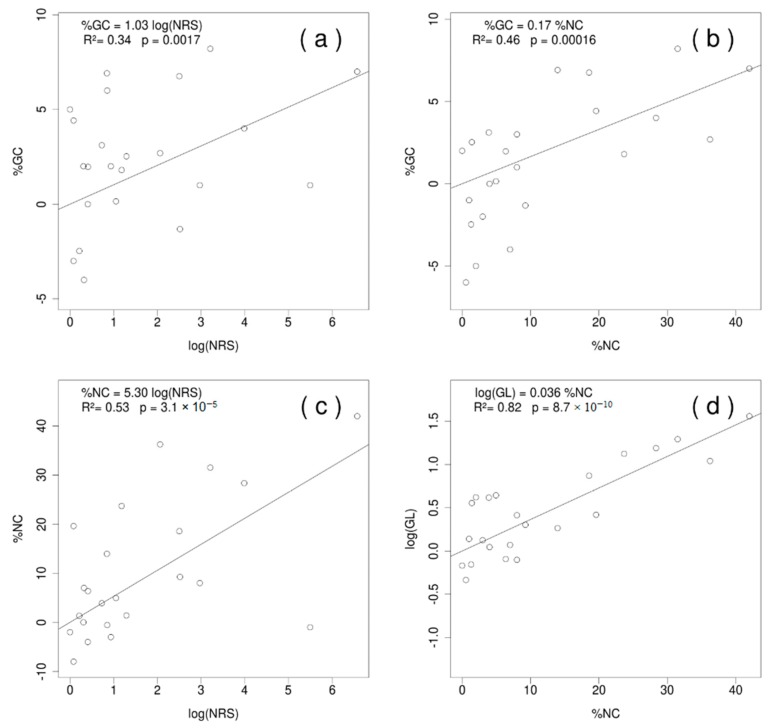
Phylogenetically independent contrasts (crunch) for the following relationships: Guanine-Cytosine (GC) content (%GC) with the number of repeated sequences (NRS) log transformed (**a**), and with the non-coding sequences (%NC) (**b**); between the %NC and NRS (**c**); and genome length (GL) with the %NC (**d**). All models were done excluding the species of Chlorophyta. For the complete set of contrasts, see the complementary material (Appendix A
Appendix A).

**Figure 3 plants-09-00358-f003:**
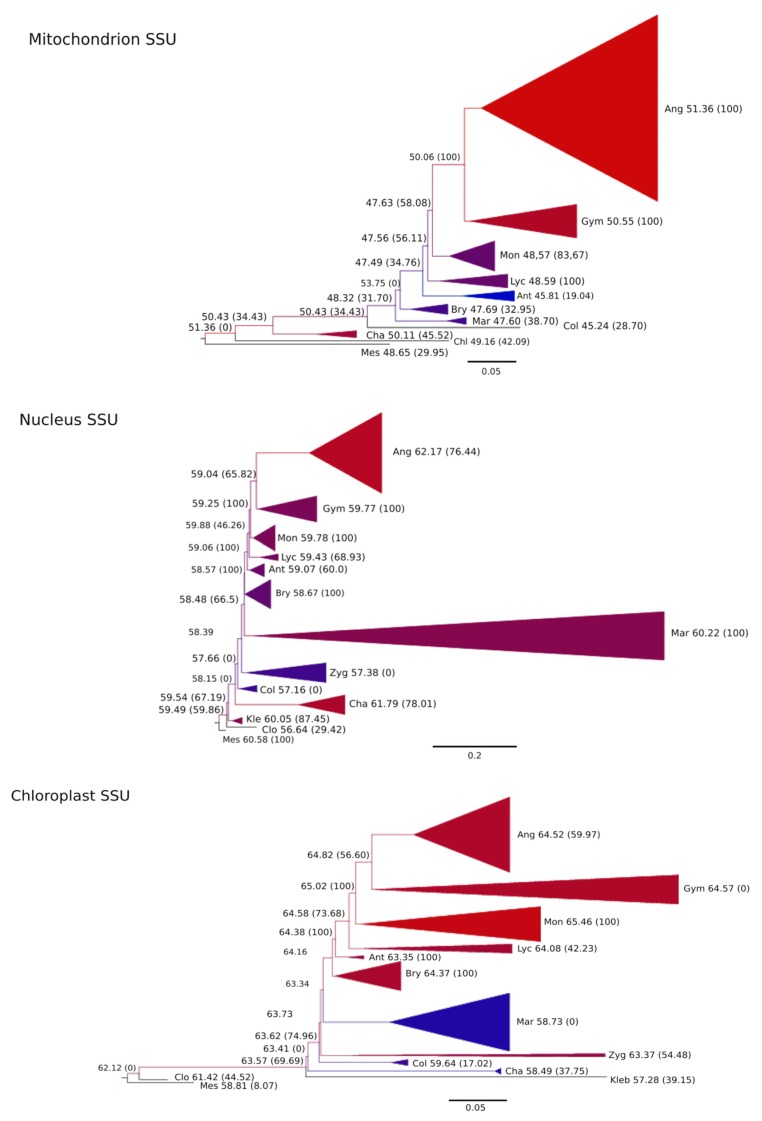
The evolution of the GC content and GC* across the phylogeny of Streptophyta: (A) Mitochondrial ribosomal Small Subunit (SSU); (B) Nuclear ribosomal SSU; (C) Plastidial ribosomal SSU. The values correspond to the ancestral GC content at the nodes, or the GC* (equilibrium GC content) in parentheses. The colors in the terminal branches represent the average GC content (blue: lowest GC content; red: highest GC content). The color scale is relative to the data set in each tree and is not directly comparable between them. The list of species and GC content at the terminal branches of each ribosomal subunit is available in Appendix A.

**Table 1 plants-09-00358-t001:** Non-homogeneous models of substitutions.

Hierarchical Models	Mitochondrion							
	Large Ribosomal Subunit (LSU)				Small Ribosomal Subunit (SSU)			
	−InL	Dev.	Df.	*P*-Value	−InL	Dev.	Df.	*P*-Value
Homogeneous	22486.50				14603.13			
NH-Model_M1^(1)^	22438.91	95.18	17	6.88 × 10^−13^	14556.56	93.14	15	2.57 × 10^−13^
NH-Model_M2^(2)^	22436.54	4.73	1	0.0296	14556.50	0.12	1	0.7234
NH-Terminal clades (as Figure 3)	22425.82	21.43	3	8.56 × 10^−5^	14548.21	16.58	3	0.0009
NH-One GC* per branch	22327.36	196.92	176	0.1337	14415.88	264.66	178	2.66 × 10^−5^
**Nucleous**								
				**Small Ribosomal Subunit (SSU)**				
					**−InL**	**Dev.**	**Df.**	***P*-Value**
Homogeneous					14023.18			
NH-Model_M1^(1)^					13984.94	76.47	19	7.47 × 10^−9^
NH-Model_M2^(2)^					13970.40	29.08	1	6.94 × 10^−8^
NH-Terminal clades (as Figure 3)					13967.78	5.23	3	0.1555
NH-One GC* per branch					13878.57	178.42	174	0.3933
**Chloroplast**								
	**Large Ribosomal Subunit (LSU)**				**Small Ribosomal Subunit (SSU)**			
	**−InL**	**Dev.**	**Df.**	***P*-Value**	**−InL**	**Dev.**	**Df.**	***P*-Value**
Homogeneous	23520.96				9895.45			
NH-Model_M1^(1)^	23324.90	392.13	19	0	9816.46	157.97	19	0
NH-Model_M2^(2)^	23323.04	3.71	1	0.0540	9816.37	0.17	1	0.6795
NH-Terminal clades (as Figure 3)	23287.39	71.31	3	2.22 × 10^−15^	9808.42	15.90	3	0.0012
NH-One GC* per branch	23083.32	408.13	174	0	9695.43	225.98	174	0.0049

^(1)^ NH-Model_M1: Non-homogeneous model which brings together to various clades; Angiosperms-Gymnosperms (seed plants) + Monilophyta-Lycopodiophyta + Bryophyta-Marchantiophyta + Anthocerotophyta + Charophyceae + Coleochaetophyceae + Zygnematophyceae + Klebsormidiophyceae + Chlorokybophyceae + Mesostigma. ^(2)^ NH-Model_M2: Like the model M1 but splitting the clades Marchantiophyta and Bryophyta. Note: -lnL: log likelihood; Dev.: residual deviance; df (degrees of freedom): residual degrees of freedom.

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
