# Peer review of "Evolutionary Trends in the Mitochondrial Genome of Archaeplastida: How Does the GC Bias Affect the Transition from Water to Land?"

_plants, 2020, doi:10.3390/plants9030358_

Round 1
Reviewer 1 Report
The article “Evolutionary trends in the mitochondrial genome of Archaeplastida: How does the GC bias affect the transition from water to land?”
is well written. It is logically divided into chapters and does not contain any grammatical mistakes.
The aim of the study is rather clear. The study has a good design. There is a large number of tables and figures of good quality presented in the manuscript.
The work has a high degree of novelty.
The manuscript can be recommended for publication in the journal after minor revision.
It is recommended to insert a list of abbreviations into the article.
It is recommended to add links to articles of 2018-2020 in chapter "References".
Author Response
Reviewer’s comment: The article “Evolutionary trends in the mitochondrial genome of Archaeplastida: How does the GC bias affect the transition from water to land?”
is well written. It is logically divided into chapters and does not contain any grammatical mistakes.
The aim of the study is rather clear. The study has a good design. There is a large number of tables and figures of good quality presented in the manuscript.
The work has a high degree of novelty.
The manuscript can be recommended for publication in the journal after minor revision.
It is recommended to insert a list of abbreviations into the article.
Author’s comment: According to the instructions for authors and one example of the article (see Yao, Y., Yang, Y., Li, C., Huang, D., Zhang, J., Wang, C., ... & Liao, W. Research progress on the functions of gasotransmitters in plant responses to abiotic stresses. Plants, 2019, 8, 605, doi:10.3390/plants8120605), we have added a list of abbreviations in the manuscript.
Reviewer’s comment: It is recommended to add links to articles of 2018-2020 in chapter "References".
Author’s comment: According to this comment, authors have included the following references (from 2018-2020):
- Kern, R.; Facchinelli, F.; Delwiche, C.; Weber, A. P.; Bauwe, H.; Hagemann, M. Evolution of photorespiratory glycolate oxidase among Archaeplastida. Plants 2020, 9,
- Wang, S.; Li, L.; Li, H.; Sahu, S.K.; Wang, H.; Xu, Y.; Xian, W.; Song, B.; Liang, H.; Cheng, S.; Chang, Y.; Song, Y.; Çebi, Z.; Wittek, S.; Reder, T.; Peterson, M.; Yang, H.; Wang, J.; Melkonian, B.; Van de Peer, Y.; Xu, X.; Ka-Shu Wong, G.; Melkonian, M.; Liu H.; Liu, X. Genomes of early-diverging streptophyte algae shed light on plant terrestrialization. Plants 2020, 6, 95-106.
- Gitzendanner, M. A.; Soltis, P. S.; Wong, G. K. S.; Ruhfel, B. R.; Soltis, D. E. Plastid phylogenomic analysis of green plants: a billion years of evolutionary history. J. Bot. 2018, 105, 291-301.
- Nishiyama, T.; Sakayama, H.; De Vries, J.; Buschmann, H.; Saint-Marcoux, D.; Ullrich, K. K.; Haas, F.B.; Vanderstraeten, L.; Becker, D.; Lang, D.; VosolsobÄ›, S.; Rombauts, S.; Wilhelmsson, P.K.I.; Janitza, P.; Kern, R.; Heyl, A.; Rümpler, F.; Villalobos, L.I.A.C.; Clay, J.M.; Skokan, R.; Toyoda, A.; Suzuki, Y.; Kagoshima, H.; Schijlen, E.; Tajeshwar, N.; Catarino, B.; Hetherington, A.J.; Saltykova, A.; Bonnot, C.; Breuninger, H.; Symeonidi, A.; Radhakrishnan, G.V.; Van Nieuwerburgh, F.; Deforce, D.; Chang, C.; Karol, K.G.; Hedrich, R.; Ulvskov, P.; Glöckner ,G.; Delwiche, C.F.; Petrášek, J.; Van de Peer, Y.; Friml, J.; Beilby, M.; Dolan, L.; Kohara, Y.; Sugano, S.; Fujiyama, A.; Delaux, P.M.; Quint, M.; Theißen, G.; Hagemann, M.; Harholt, J.; Dunand, C.; Zachgo, S.; Langdale, J.; Maumus, F.; Van Der Straeten, D.; Gould, S.B.; Rensing, S.A. The Chara genome: secondary complexity and implications for plant terrestrialization. Cell 2018, 174, 448-464.
- Chaitanya, K. V. Orgenellar Genome Analysis. In Genome and Genomics. Springer, Singapore, 2019, pp. 89-119.
- Turmel, M.; Lemieux, C. Evolution of the plastid genome in green algae. In Advances in Botanical Research. Academic Press, New York, USA, 2018, Vol. 85, pp. 157-193.
- Zheng, F.; Wang, B.; Shen, Z.; Wang, Z.; Wang, W.; Liu, H.; Wang, C.; Xin, M. The chloroplast genome sequence of the green macroalga Caulerpa okamurae (Ulvophyceae, Chlorophyta): Its structural features, organization and phylogenetic analysis. Genomics 2020, 100752.

Reviewer 2 Report
The transition from water to land in Archeaplastida was accompagnied with the gain of several functions such as change in sexuality ..., as fruit and seed complexity. Subsequent changes in the 3 genomes were statistically documented here, interestingly. Different models for genome evolution in the mitochondria and the plastids could explain most features. Basing on correlation between parameters (GL, %NC, %GC, NRS, and RSL) Authors have made constats that should be published. The diversity of the populations parameters and biology between these taxa leaves open several explanations to explain increase in genome size, in the nucleus and in the mitochondria. In higher plants the plastid DNA is reduced by deletions in parasitic plants Philipanche and Orobanche, that sustains its major importance in all taxa studied since it remains stable. I agree that %GC change is a subsequent variation due to strong selection rather than gBGC.
I just have a suggestion to be accurate. Genome size should be split into haploid set (true genome length) and amphiploid set (several haploid genome together). Plants have conquered new spaces because of archaeopolyploid events gathering complementary adaptative functions in one individual.
excellent work
Author Response
Reviewer’s comment: The transition from water to land in Archeaplastida was accompagnied with the gain of several functions such as change in sexuality ..., as fruit and seed complexity. Subsequent changes in the 3 genomes were statistically documented here, interestingly. Different models for genome evolution in the mitochondria and the plastids could explain most features. Basing on correlation between parameters (GL, %NC, %GC, NRS, and RSL) Authors have made constats that should be published. The diversity of the populations parameters and biology between these taxa leaves open several explanations to explain increase in genome size, in the nucleus and in the mitochondria. In higher plants the plastid DNA is reduced by deletions in parasitic plants Philipanche and Orobanche, that sustains its major importance in all taxa studied since it remains stable. I agree that %GC change is a subsequent variation due to strong selection rather than gBGC.
I just have a suggestion to be accurate. Genome size should be split into haploid set (true genome length) and amphiploid set (several haploid genome together). Plants have conquered new spaces because of archaeopolyploid events gathering complementary adaptative functions in one individual.
excellent work
Author’s comment: Thank you very much for your comment. Your reflections guarantee the validity of our data.
